# Modeling Acoustic Emission Due to an Internal Point Source in Circular Cylindrical Structures

Kwang Bok Kim, Bong Ki Kim and Jun-Gill Kang *

Rm #306, Integrity Diagnostics Korea (IDK), IT Venture Town, 35, Techno 9-ro, Yuseong-gu, Daejeon 34027, Republic of Korea
* Correspondence: jgkang@cnu.ac.kr

**Abstract:** In one of our previous publications, we developed the first mathematical model for acoustic emission from an internal point source in a transversely isotropic cylinder. The point source, as an internal defect, is the most fundamental source generating AE in homogeneous media; it is represented by a spatiotemporal concentrated force and generates three scalar potentials for compressional, and horizontally and vertically polarized shear waves. The mathematical formulas for the displacements were derived by introducing the concentrated force-incorporated potentials into the Navier–Lamé equation. Since the publication of that paper, we detected some errors. In this paper, we correct the errors and extend the analytical modeling to a cylindrical shell structure. For acoustic emission in general circular cylindrical structures, we derived solutions by applying the boundary conditions at inner and outer surfaces of the structures. Under these conditions, we solve the radial, tangential, and axial displacement fields. Analytical simulations of the acoustic emission were carried out at several point source locations for circular cylindrical geometries. We show that the maximum amplitude of the axial displacement is dependent on the point source position and $2\pi$-aperiodicity of the cylindrical geometry. Our mathematical formulas are very useful for characterizing AE features generated from an internal defect source in cylindrical geometries.

**Keywords:** acoustic emission; concentrated force-incorporated potentials; cylindrical structures; mathematical formula

## 1. Introduction

Many conventional techniques applied in nondestructive testing are based on an active mode, in which testing loadings are applied during testing to deliver signals or energy from the outside to the test body. In contrast, the acoustic emission (AE) technique is a passive method that does not require the application of external energy to the test structure, as AE is generated by a material as a result of a sudden release of energy (other than heat) from localized sources within the solid, in turn due to a failure of lattice vibrations in materials. AE sources are usually classified as primary or secondary. Primary AE sources include material degradations related to deformation and fracture development, whereas leak, flow, and the fabrication process are secondary AE sources. Due to this unique characteristic, AE techniques are uniquely applicable to structure health monitoring (SHM) [1–4].

Among the primary AE sources, crack formation and growth are the most important practical nondestructive testing (NDT) because the detecting and monitoring of these failures can prevent or slow further damage. Thus, in SHM, the point source (PS) is adapted as an AE excitation source; for example in seismic displacements, crack fracture and cleavage, and concentrated vertical step forces [5–10]. The displacements generated by a PS were first introduced by Stokes [11]. Later, Achenbach presented mathematical formulas for displacements in spherical geometry by defining it as a concentrated force (CF) loaded at a point [12]. In mathematical formulas, Helmholtz potentials for the displacements were derived in terms of scalar potentials generated by PS excitation in an infinite domain [12–14]. Although the AE generated by the PS is important for characterizing real signals observed

in practical SHM, theoretical modeling was limited to spherical geometries in an infinite domain. To our knowledge, no theoretical work on AE excited by an internal PS in cylindrical geometries was reported in the literature. Most theoretical works on elastic wave properties in cylindrical geometries focused only on wave propagation and scattering situations with or without external perturbations [15–23].

In linear elastodynamics, the Navier–Lamé (NL) equation is the most popular and effective method for solving the displacement fields in elastic media. In cylindrical coordinates, the NL equation was solved by combining three potentials responsible for one compression (P) wave and two shear waves (horizontally and vertically polarized; SH and SV, respectively), using the models proposed by Morse and Feshbach [24] and Buchwald [25], respectively. The basic difference between the two models is that the compression and shear parts are separated in one of them, but not the other. The two models were examined comprehensively for the case with no body forces by Honarvar et al. [15,18] and Sakhr et al. [20,21]. Additionally, Shatalov et al. developed a method to find exact solutions for axisymmetric wave propagation in functional cylinders by matching continuity and boundary conditions at layer junctions [26,27].

Previously, we constructed the NL equation specifically for displacements generated by PS excitation in a transversely isotropic cylinder (TIC) [28] and a two-dimensional plate [29]. As an internal defect, the CF acting in a preferent direction had both spatial and temporal properties, represented by the delta function and time-dependent CF vector. As the first task, Green's function for the delta function was determined in a given geometry. Green's function contributes the CF distribution in a given domain. For TIC, the CF vector contributes to the temporal Helmholtz potentials for P, SV, and SH waves based on the Morse and Feshbach model. Contrary to ordinary potentials, these potentials are characterized by the nature of the material defect. For discriminating from ordinary ones, these potentials are referred to as CF-incorporated potentials (CFIPs), which reduce the original scalar components of the NL equation into three independent partial differential equations (PDEs) for P, SV, and SH waves. The exact solution was solved and applied to the P, SV, and SH displacement simulations.

The main purpose of this work is to present a completeness theorem applicable to AE due to CF excitation in circular cylindrical geometries, including shell and rod configurations. In Ref. [28], we confined the displacement solutions with a $2\pi$-periodic (azimuthally free) angular part in the tangential component. We reconstructed the displacement solutions by applying a fundamental set of free-surface conditions on outer and/or inner circumferences and $2\pi$-aperiodicity with azimuthal functions. Absolute values of the displacement fields were calculated at a given point on the outer surface, in which two modes of the CF along the radial and axial directions were considered. To our knowledge, the proposed method is the only one in the literature that is fully applicable to cylindrical geometries. The analytical modeling presented in this paper provides insight into the overall AE signal process from generation and propagation to reception. We offer a systematic and unifying solution method that can be used to evaluate the AE signal generated from an internal crack in a cylindrical structure.

## 2. Green's Function

In Ref. [28], the CF as the internal PS located at $x_0$ is formulated in terms of an oscillating impulse with natural frequencies of the material in a given geometry, as follows:

$$\boldsymbol{f} = \boldsymbol{P}\delta(\boldsymbol{x} - \boldsymbol{x}_0)e^{-i\omega t} \tag{1}$$

where $\boldsymbol{P}$ is the force vector and $\omega$ is the predominant angular frequency ($\omega = 2\pi\nu$) of AE. Dirac's delta function $\delta(\boldsymbol{x} - \boldsymbol{x}_0)$ provides a method for solving spatial problems dealing with a PS. Green's function $g(\boldsymbol{x}; \boldsymbol{x}_0)$, defined as

$$\nabla^2 g(\boldsymbol{x}; \boldsymbol{x}_0) = \delta(\boldsymbol{x} - \boldsymbol{x}_0) \tag{2}$$

Is the solution for the delta function in elastodynamics, and provides the spatial distribution of the CF at a given time. In cylindrical coordinates, Equation (2) is expressed as

$$\nabla^2[g_r(r;r_0)g_\theta(\theta;\theta_0)g_z(z;z_0)] = \frac{\delta(r-r_0)\delta(\theta-\theta_0)\delta(z-z_0)}{r} \quad (3)$$

where $\nabla^2$ is the Laplacian in cylindrical coordinates, and the CF is located at $r_0, \theta_0$, and $z_0$. At any point in the cylindrical domain except the CF locating point, Equation (3) becomes zero,

$$\nabla^2[g_r(r;r_0)g_\theta(\theta;\theta_0)g_z(z;z_0)] = 0. \quad (4)$$

In cylindrical $(r, \theta, z)$ coordinates, Equation (4) can be rewritten as

$$\frac{r^2}{g_r}\frac{\partial^2 g_r}{\partial r^2} + \frac{r}{g_r}\frac{\partial g_r}{\partial r} + \frac{1}{g_\theta}\frac{\partial^2 g_\theta}{\partial \theta^2} + \frac{r^2}{g_z}\frac{\partial^2 g_z}{\partial z^2} = 0.$$

Substituting the following relations

$$\frac{1}{g_z}\frac{\partial^2 g_z}{\partial z^2} = \kappa_z^2,$$

$$\frac{1}{g_\theta}\frac{\partial^2 g_\theta}{\partial \theta^2} = -v^2,$$

Into the above PDE results, we obtain

$$r^2\frac{\partial^2 g_r}{\partial r^2} + r\frac{\partial g_r}{\partial r} + \left(\kappa_z^2 r^2 - v^2\right)g_r = 0$$

Note that we selected exponential, rather than oscillating, solutions in the z-direction. This implies that the radial solutions are appropriate for the particular set of boundary conditions under consideration. In ref. [28], we obtained the axial, tangential, and radial components of Green's function under certain conditions, such as the continuity and discontinuity principles, angular symmetry for the cylindrical domain with length $l$ and a radius of $a$. The inner radius $b$ is introduced to the shell structure as an additional radial boundary condition.

$$g_z(z;z_0) = \begin{cases} -\frac{1}{2\kappa_z}e^{-\kappa_z(z_0-z)} & 0 < z < z_0 < l \\ -\frac{1}{2\kappa_z}e^{-\kappa_z(z-z_0)} & 0 < z_0 < z < l \end{cases} \quad (5)$$

$$g_\theta(\theta;\theta_0) = \cos[v(\theta-\theta_0)] \ (v = 0, \pm 1, \pm 2, \cdots), \quad (6)$$

$$g_r(r;r_0) = J_v(\kappa_z r_0)J_v(\kappa_z r) \quad (b < r < a) \quad (7)$$

In Equation (7), $J_v$ is a Bessel function of the first kind of $v$-th order. The Bessel function of the second kind is excluded because there is no singularity in the cylindrical domain. The value of $\kappa_z$ can be obtained by applying the boundary conditions at the outer or inner surfaces of the cylindrical shell to Equation (7):

$$J_v(\kappa_z r)|_{r=a-r_0} = 0 \ (b < r_0 < r < a),$$

$$J_v(\kappa_z r)|_{r=r_0-b} = 0 \ (b < r < r_0 < a).$$

As an alternative, the boundary condition $g_r(r;r_0)|_{r=a} = 0$ was applied [30]. However, this condition resulted in a discontinuous and asymmetric Green's function. Selecting the first root $(r_{v1})$ of the Bessel function, $r_{v1} = \kappa_z(a - r_0)$ or $r_{v1} = \kappa_z(r_0 - b)$. These relations give

$$\kappa_z = \begin{cases} \frac{r_{v1}}{(a-r_0)} & (a+b > 2r_0) \\ \frac{r_{v1}}{(r_0-b)} & (a+b < 2r_0) \end{cases}. \quad (8)$$

For the outer surface, $\kappa_z = \frac{r_{v1}}{(a-r_0)}$. Introducing a parameter, $A_{v1}$, we obtain Green's function:

$$g(r, \theta, z; r_0, \theta_0, z_0) = A_{v1} \, J_v(\kappa_z r_0) J_v(\kappa_z r) \cos[v(\theta - \theta_0)]$$
$$\times \begin{cases} -\frac{1}{2\kappa_z} e^{-\kappa_z(z_0-z)} & (0 < z < z_0 < l) \\ -\frac{1}{2\kappa_z} e^{-\kappa_z(z-z_0)} & (0 < z_0 < z < l) \end{cases} . \tag{9}$$

In ref. [28], the parameter $A_{v1}$ was determined by integrating the delta function in the cylindrical domain. The integrations of $\delta(\theta - \theta_0)\delta(z - zs_0)$ for the cylindrical shell are the same as those for the cylinder, because these two delta functions are independent of $r$. The remaining problem is to integrate $\frac{\delta(r-r_0)}{r}$,

$$\pi A_{v1} J_v(\kappa_z r_0) J_v(\kappa_z r) \left[ e^{-\kappa_z z_0} + e^{-\kappa_z(l-z_0)} - 2 \right] = \frac{\delta(r - r_0)}{r}. \tag{10}$$

Multiplying both sides of Equation (10) by $J_p(\kappa_z r) \, rdr$ and integrating over $(b, a)$ gives the following:

$$\pi A_{v1} \left[ e^{-\kappa_z z_0} + e^{-\kappa_z(l-z_0)} - 2 \right] J_v(\kappa_z r_0) \int_b^a J_v(\kappa_z r) J_p(\kappa_z r) rdr$$
$$= \int_b^a \delta(r - r_0) J_p(\kappa_z r) \, dr. \tag{11}$$

The integrations on the left side of the above equation can be divided into two parts:

$$\int_b^a J_v(\kappa_z r) J_p(\kappa_z r) rdr = \left\{ \int_{r_0}^a [J_v(\kappa_z r)]^2 rdr + \int_b^{r_0} [J_v(\kappa_z r)]^2 rdr \right\} \delta_{pv}.$$

Applying the normalization and orthogonality principles of Bessel functions,

$$\int_0^c J_p\left(\alpha \frac{r}{c}\right) J_q\left(\alpha \frac{r}{c}\right) rdr = \frac{c^2}{2} [J_{p+1}(\alpha)]^2 \delta_{pq},$$

And the following concepts of the delta function,

$$\int_b^a J_v(\kappa_z r) J_p(\kappa_z r) = \frac{a^2}{2} [J_{v+1}(\kappa_z a)]^2 - \frac{b^2}{2} [J_{v+1}(\kappa_z b)]^2$$

$$\int_b^a \delta(r - r_0) J_v(\kappa_z r) \, dr = J_v(\kappa_z r_0),$$

To Equation (11), we obtain the constant

$$A_{v1} = \frac{1}{\pi \left[ e^{-\kappa_z z_0} + e^{-\kappa_z(l-z_0)} - 2 \right]} \times \frac{2}{\left\{ a^2 [J_{v+1}(\kappa_z a)]^2 - b^2 [J_{v+1}(\kappa_z b)]^2 \right\}}. \tag{12}$$

For the cylindrical problem, the corrected value of $A_{v1}$ is given in (A1). Introducing Equation (12) to Equation (9) gives the Green's function for the Kronecker delta function:

$$g(r, \theta, z; r_0, \theta_0, z_0) = G_{v1} \, J_v(\kappa_z r_0) J_v(\kappa_z r) \{ \cos[v(\theta - \theta_0)] \}, \tag{13}$$

where

$$G_{v1} = -\frac{1}{2\kappa_z} A_{v1} J_v(\kappa_z r_0) \times \begin{cases} e^{-\kappa_z(z_0-z)} & (0 < z < z_0 < l) \\ e^{-\kappa_z(z-z_0)} & (0 < z_0 < z < l) \end{cases} . \tag{14}$$

Since the CF direction is determined by the $P$ vector, Green's function is assumed to be azimuthally independent ($v = 0$). In addition, it is very convenient to take the location of the PS as the new origin and introduce relative coordinates to the location of the PS, defined as $\xi = r - r_0$, $\vartheta = \theta - \theta_0$, and $\eta = z - z_0$, where $\xi_0 = 0$, $\vartheta_0 = 0$, and $\eta_0 = 0$ (hereafter,

referred to as the PS-oriented $(\xi, \vartheta, \eta)$ coordinate system), Equations (13) and (14) can be rewritten as

$$g(\xi, \eta) = G_{01} J_0(\kappa_z \xi), \tag{15}$$

$$G_{01} = -\frac{1}{2\kappa_z} A_{01} J_0(\kappa_z \xi_0) \times \begin{cases} e^{\kappa_z \eta} & (z_0 - l < \eta < 0) \\ e^{-\kappa_z \eta} & (0 < \eta < l - z_0) \end{cases}. \tag{16}$$

In Equation (15), the value of $\xi$ at a given point is the shortest distance from the PS. In cylindrical geometry,

$$\xi = \sqrt{(x_i - x_{0i})^2 + (x_j - x_{0j})^2} \rightarrow \left( r_0 = \sqrt{x_{0i}^2 + x_{0j}^2} \right). \tag{17}$$

For the shell geometry, the calculation of $\xi$ is somewhat complicated due to its hollow interior. No linear distance exists between the two points across the hollow interior. In Figure 1b, an arc connecting the PS and a given point is introduced. This connection should not intersect with the hollow circle. The arc rises at a constant rate between $r_0$ and ($r_p < r_0$), given as $R = \frac{r_P - r_0}{\theta}$ $\left( r_0 = \sqrt{x_{0i}^2 + x_{0j}^2}, \; r_P = \sqrt{x_i^2 + x_j^2} \right)$, where $\theta$ is the angle between the PS and a point P. If the angle, $d\theta$, between $r_1$ and $r_2$ is infinitesimal, the arc length, $d\varsigma$, connecting the two points, becomes a line. Applying the cosine rule to the triangle,

$$d\varsigma = \sqrt{r_1^2 + r_2^2 - 2r_1 r_2 \cos d\theta}$$
$$= \sqrt{r_1^2 + (r_1 + R d\theta)^2 - 2r_1(r_1 + R d\theta) \cos(d\theta)}.$$

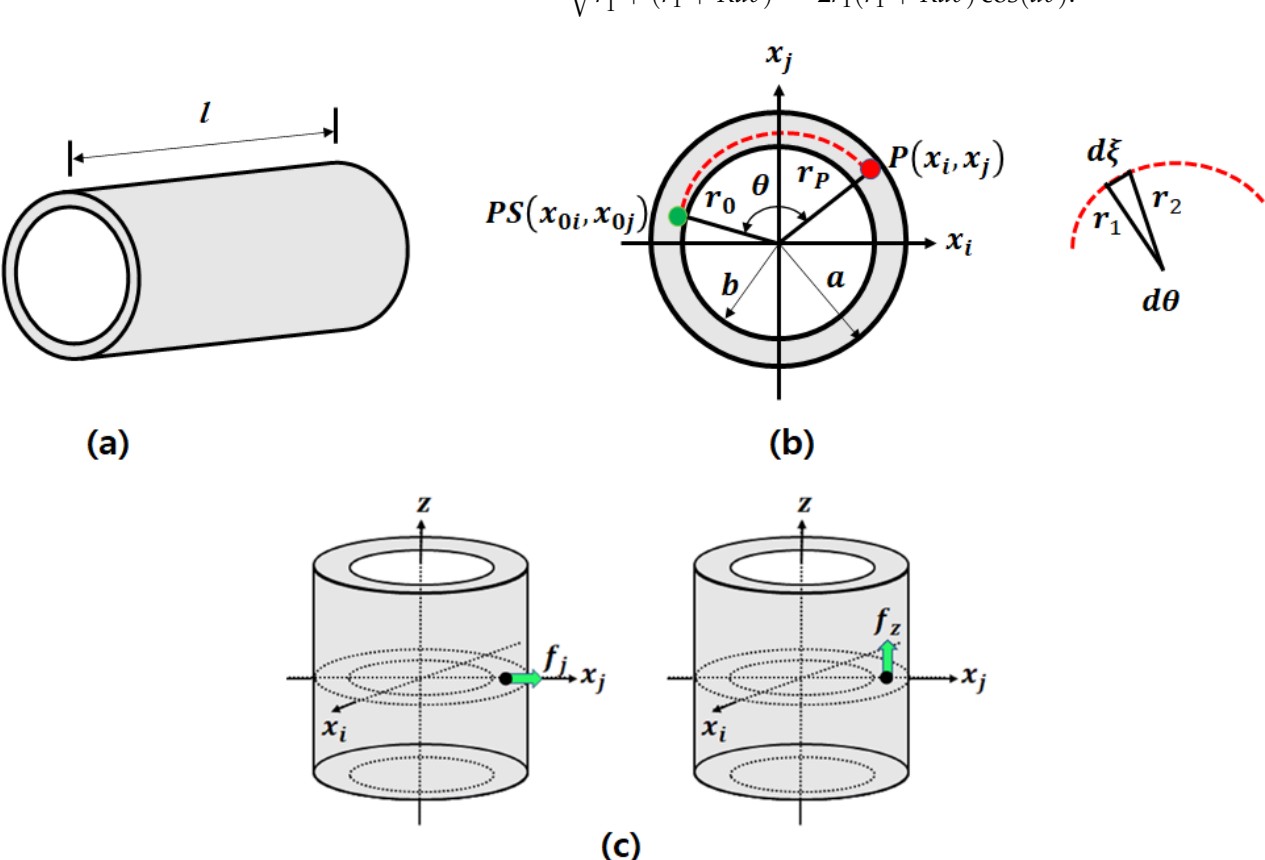

**Figure 1.** (**a**) Geometry of a thick cylindrical shell, (**b**) its radial cross-section involving the point source (PS) and a given point P, in which the red dotted line represents an arc connecting PS and P, and (**c**) two forms of the PS vector along the $x_j$ and z directions used in analytical modeling.

Assuming that $\cos(d\theta) \approx 1 - \frac{(d\theta)^2}{2}$ and $(d\theta)^3 \approx 0$, the above equation becomes

$$
\begin{aligned}
\varsigma &= \int_{\theta=0}^{\theta=\theta} \sqrt{R^2 + r_0^2}\, d\theta \\
&= \sqrt{r_0^2 + \left(\frac{r_P - r_0}{\theta}\right)^2}\, \theta \rightarrow \left[\theta = \arctan\left(\frac{x_j}{x_i}\right) - \arctan\left(\frac{x_{0j}}{x_{0i}}\right)\right].
\end{aligned}
\tag{18}
$$

Figure 2 shows the Green's function calculated for the cylinder ($a = 0.5$ m, $b = 0$) and shell ($a = 0.5$ m, $b = 0.4$ m). It can be shown that the calculated functions are continuous and symmetric with respect to the PS location.

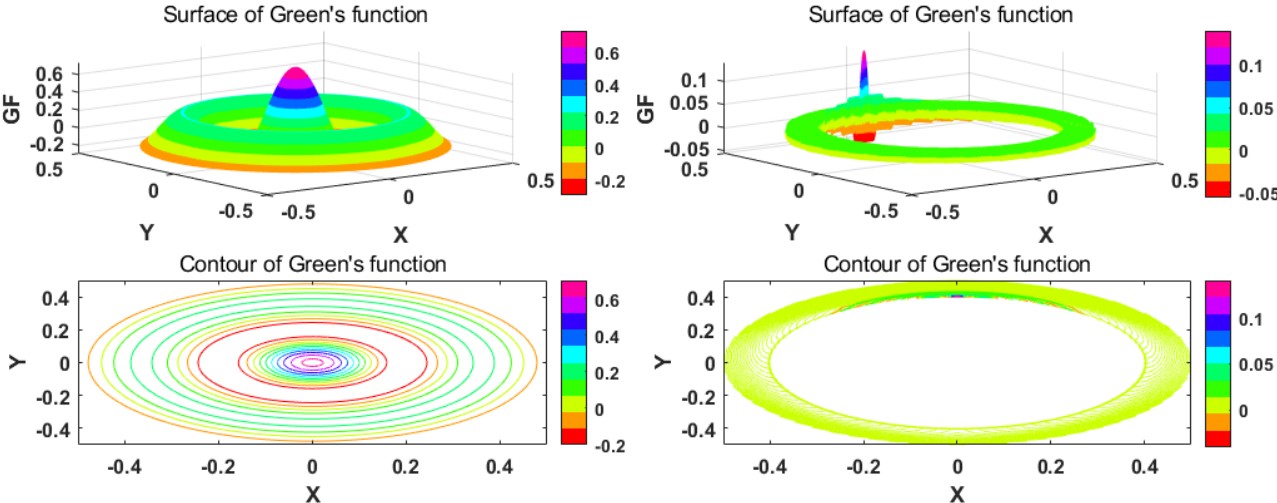

**Figure 2.** Calculated Green's function for: (a) a cylinder and (b) a shell, in which the PS locates at $\left(x_{0i} = 0, x_{0j} = 0\right)$ and $\left(x_{0i} = 0, x_{0j} = 0.45\right)$, respectively. The plots are conducted by applying [th, r] = meshgrid(linspace(0, 2*pi, 2000), linspace(b, a, 2000)) and [X, Y] = pol2cart(th, r) in MATLAB® (MathWorks, Natick, MA, USA). The peak position in [X, Y] corresponds to the $(x_{0,i}, x_{0j})$ Cartesian coordinates.

## 3. Displacement Fields Generated by a Point Source

The NL equation for the displacement field $\boldsymbol{u}$ in an elastic and homogeneous medium subject to a local body force $\boldsymbol{f}$ is given by [31]

$$
(\lambda + 2\mu)\nabla(\nabla\cdot\boldsymbol{u}) - \mu\nabla \times (\nabla \times \boldsymbol{u}) + \boldsymbol{f} = \rho\frac{\partial^2 \boldsymbol{u}}{\partial t^2},
\tag{19}
$$

where $\lambda$ and $\mu$ are the Lamé constants, and $\rho$ is the density of the media. The displacement field $\boldsymbol{u}$ in cylindrical coordinates proposed by Morse and Freshbach involves three potential functions $\Phi$ for the P wave, $X\hat{\boldsymbol{e}}_z$ for the SH wave, and $\Psi\hat{\boldsymbol{e}}_z$ for the SV wave. The representation of $\boldsymbol{u}$ for cylindrical geometries is given by

$$
\boldsymbol{u} = \nabla\Phi + \nabla \times (X\hat{\boldsymbol{e}}_z) + a\nabla \times \nabla \times (\Psi\hat{\boldsymbol{e}}_z),
\tag{20}
$$

where $a$ is the radius of the cylinder. For the displacement field generated by the force vector $\boldsymbol{P}$ due to an intrinsic point defect, the three potential functions $\Phi$, $X$, and $\Psi$ are correlated with the force vector $\boldsymbol{P}$ (referred to as CFIPs). CFIPs in the PS-oriented $\left(\xi_i, \xi_j, \eta\right)$ Cartesian coordinates are defined as

$$
\Phi = \nabla\cdot\boldsymbol{P}\phi = \frac{\partial(\boldsymbol{P}\phi)}{\partial\xi_i} + \frac{\partial(\boldsymbol{P}\phi)}{\partial\xi_j} + \frac{\partial(\boldsymbol{P}\phi)}{\partial\eta},
\tag{21}
$$

$$X\hat{e}_z = -\nabla \times P\chi = -\left[\left(\frac{\partial P\chi}{\partial \xi_i}\right) - \left(\frac{\partial P\chi}{\partial \xi_j}\right)\right]\hat{e}_z, \tag{22}$$

$$\Psi\hat{e}_z = \nabla \times P\psi = \left[\left(\frac{\partial P\psi}{\partial \xi_i}\right) - \left(\frac{\partial P\psi}{\partial \xi_j}\right)\right]\hat{e}_z, \tag{23}$$

where $\phi$, $\chi$, and $\psi$ are scalar functions for P, SH, and SV waves, respectively. These scalar functions are expressed by

$$\phi(\xi_i, \xi_j, \eta, t) = \phi(\xi_i, \xi_j, \eta)e^{-i\omega t},$$

$$\chi(\xi_i, \xi_j, \eta, t) = \chi(\xi_i, \xi_j, \eta)e^{-i\omega t},$$

$$\psi(\xi_i, \xi_j, \eta, t) = \psi(\xi_i, \xi_j, \eta)e^{-i\omega t}.$$

Combining Equations (19)–(23) with Equations (1) and (2), and replacing the Lamé constants and $\rho$ by the longitudinal wave speed $\left(c_P = \sqrt{\lambda + 2\mu/\rho}\right)$ and transverse wave speed $\left(c_S = \sqrt{\mu/\rho}\right)$ leads to

$$\nabla^2\phi + k_p^2\phi + \frac{k_p^2}{\rho\omega^2}g = 0, \tag{24}$$

$$\nabla^2\chi + k_s^2\chi + \frac{k_s^2}{\rho\omega^2}g = 0, \tag{25}$$

$$\nabla^2\psi + k_s^2\psi = 0, \tag{26}$$

where $k_p = \frac{\omega}{c_p}$ and $k_s = \frac{\omega}{c_s}$, corresponding to the angular wavenumbers of the P and the S waves, respectively. The solutions of these PDEs in the PS-oriented $(\xi, \vartheta, \eta)$ cylindrical coordinates are as follows:

$$\phi(\xi, \vartheta, \eta) = A_m J_m(\alpha\xi)J_m(\alpha\xi)\cos(m\vartheta)e^{-ik_\eta\eta} - \frac{k_p^2}{\rho\omega^2}\frac{G_{01}}{\alpha^2 - \kappa_z^2}J_m(\kappa_z\xi), \tag{27}$$

$$\chi(\xi, \vartheta, \eta) = B_m J_m(\beta\xi)\sin(m\vartheta)e^{-ik_\eta\eta} - \frac{k_p^2}{\rho\omega^2}\frac{G_{01}}{\beta^2 - \kappa_z^2}J_m(\kappa_z\xi), \tag{28}$$

$$\psi(\xi, \vartheta, \eta) = C_m J_m(\beta\xi)\cos(m\vartheta)e^{-ik_\eta\eta}, \tag{29}$$

where $\alpha^2 = k_p^2 - k_\eta^2$ and $\beta^2 = k_s^2 - k_\eta^2$. The corrections for the particular solutions in Ref. [28] are given in Appendix A.

In the same manner as in ref. [28], the force vectors $P$s acting in the radial $(P_j)$ and the axial $(P_z)$ directions are introduced to solve for $\Phi$, $X\hat{e}_z$, and $\Psi\hat{e}_z$. By substituting Equation (27) into Equation (21) with coordination conversion, we obtain the CFIP for the P wave

$$\Phi_j = P_j\Xi\left[A_{mj}\frac{\partial J_m(\alpha\xi)}{\partial \xi}\cos(m\vartheta)e^{-ik_\eta\eta} - \frac{k_p^2}{\rho\omega^2}\frac{G_{01}}{\alpha^2 - \kappa_z^2}\frac{\partial J_m(\kappa_z\xi)}{\partial \xi}\right], \tag{30}$$

$$\Phi_z = P_z\left[(-ik_\eta)A_{mz}J_m(\alpha\xi)\cos(m\vartheta)e^{-ik_\eta\eta} - \frac{k_p^2}{\rho\omega^2}\frac{\partial G_{01}}{\partial \eta}\frac{1}{\alpha^2 - \kappa_z^2}J_m(\kappa_z\xi)\right]. \tag{31}$$

In Equation (31),

$$\Xi = \frac{\xi_i}{\xi}\delta_{ij} + \frac{\xi_j}{\xi} = \frac{a\cos\varphi - x_{0i}}{\sqrt{\xi_i^2 + \xi_j^2}}\delta_{ij} + \frac{a\sin\varphi - x_{0j}}{\sqrt{\xi_i^2 + \xi_j^2}}, \tag{32}$$

where $\varphi$ is the angle between an observation point and the $x_i$ axis. Similarly, we obtain the CFIP for the SH wave

$$X_j = -P_j\Sigma\left[B_{mj}\frac{\partial J_m(\beta\xi)}{\partial\xi}\ \sin(m\vartheta)e^{-ik_\eta\eta} - \frac{k_p^2}{\rho\omega^2}\frac{G_{01}}{\beta^2-\kappa_z^2}\frac{\partial J_m(\kappa_z\xi)}{\partial\xi}\right], \tag{33}$$

$$X_z = -P_z\left(\frac{\partial\chi}{\partial\xi_i}\delta_{iz} - \frac{\partial\chi}{\partial\xi_j}\delta_{jz}\right) = 0, \tag{34}$$

where

$$\Sigma = \frac{\xi_i}{\xi} - \frac{\xi_j}{\xi}\delta_{ij} = \frac{a\cos\varphi - x_{0i}}{\sqrt{\xi_i^2 + \xi_j^2}} - \frac{a\sin\varphi - x_{0j}}{\xi}\delta_{ij}. \tag{35}$$

The CFIP for the SV wave is given by

$$\Psi_j = -P_j\Sigma C_{mj}\frac{\partial J_m(\beta\xi)}{\partial\xi}\ \cos(m\vartheta)e^{-ik_\eta\eta}, \tag{36}$$

$$\Psi_z = \left[P_j\left(\frac{\partial\psi}{\partial\xi_i}\right) - P_i\left(\frac{\partial\psi}{\partial\xi_j}\right)\right] = 0. \tag{37}$$

All three CFIPs for the P, SH, and SV waves were completely determined in the given CF direction. In ref. [28], detailed derivation of the displacement components for the cylindrical geometry are described in terms of $\Phi$, $X$, and $\Psi$. For the cylindrical geometries, applying gradient, divergence, and curl operators, Equation (20) results in the displacement components in the $(r, \theta, z; r_0, \theta_0, z_0)$ coordinates as

$$\boldsymbol{u} = u_r\hat{\boldsymbol{r}} + u_\theta\hat{\boldsymbol{\theta}} + u_z\hat{\boldsymbol{z}},$$

where $u_r, u_\theta,$ and $u_z$ are the radial, tangential, and axial displacements, respectively. In the $(\xi, \vartheta, \eta)$ coordinates, these displacement components are as follows:

$$u_r = \frac{\partial\Phi}{\partial\xi} + \frac{1}{\xi}\frac{\partial X}{\partial\vartheta} + a\frac{\partial^2\Psi}{\partial\xi\partial\eta}, \tag{38}$$

$$u_\theta = \frac{1}{\xi}\frac{\partial\Phi}{\partial\vartheta} - \frac{\partial X}{\partial\xi} + \frac{a}{\xi}\frac{\partial^2\Psi}{\partial\vartheta\partial\eta}, \tag{39}$$

$$u_z = \frac{\partial\Phi}{\partial\eta} - a\left(\frac{\partial^2\Psi}{\partial\xi^2} + \frac{1}{\xi}\frac{\partial\Psi}{\partial\xi} + \frac{1}{\xi^2}\frac{\partial^2\Psi}{\partial\vartheta^2}\right). \tag{40}$$

Substituting Equations (30), (33), and (36) into Equations (38)–(40) gives the displacement $d$ component due to $P_j$ as

$$u_{dj} = P_j\left(A_{mj}F_{dj}^1 + B_{mj}F_{dj}^2 + C_{mj}F_{dj}^3 + F_{dj}^4\right)e^{-i\omega t}. \tag{41}$$

Notably, the component $F_{df}^4$ ($f = j$ or $z$) is obtained from the particular solutions for P and SH potentials associated with Green's function. In this paper, the components $F_{df}^4$ in ref. [28] is corrected.

For the radial component $u_{rj}$,

$$F_{rj}^1 = \Xi\frac{\partial^2 J_m(\alpha\xi)}{\partial\xi^2}\cos(m\vartheta)e^{-ik_\eta\eta}, \tag{42}$$

$$F_{rj}^2 = \frac{m\Sigma}{\xi}\frac{\partial J_m(\beta\xi)}{\partial\xi}\cos(m\vartheta)e^{-ik_\eta\eta}, \tag{43}$$

$$F_{rj}^3 = ik_\eta a\Sigma \frac{\partial^2 J_m(\beta\xi)}{\partial\xi^2} \cos(m\vartheta)e^{-ik_\eta\eta}, \tag{44}$$

$$F_{rj}^4 = -\Xi \left(\frac{k_p^2}{\rho\omega^2}\right) G_{01} \frac{1}{\alpha^2 - \kappa_z^2} \frac{\partial^2 J_m(\kappa_z\xi)}{\partial\xi^2}. \tag{45}$$

For the tangential component $u_{\theta j}$

$$F_{\theta j}^1 = -\frac{m\Xi}{\xi} \frac{\partial J_m(\alpha\xi)}{\partial\xi} \sin(m\vartheta)e^{-ik_\eta\eta}, \tag{46}$$

$$F_{\theta j}^2 = -\Sigma \frac{\partial^2 J_m(\beta\xi)}{\partial\xi^2} \sin(m\vartheta)e^{-ik_\eta\eta}, \tag{47}$$

$$F_{\theta j}^3 = -\frac{ik_\eta am\Sigma}{\xi} \frac{\partial J_m(\beta\xi)}{\partial\xi} \sin(m\vartheta)e^{-ik_\eta\eta}, \tag{48}$$

$$F_{\theta j}^4 = -\Sigma \left(\frac{k_s^2}{\rho\omega^2}\right) G_{01} \frac{1}{\beta^2 - \kappa_z^2} \frac{\partial^2 J_m(\kappa_z\xi)}{\partial\xi^2}. \tag{49}$$

For the axial component $u_{zj}$,

$$F_{zj}^1 = -ik_\eta \Xi \frac{\partial J_m(\alpha\xi)}{\partial\xi} \cos(m\vartheta)\, e^{-ik_\eta\eta}, \tag{50}$$

$$F_{zj}^2 = 0, \tag{51}$$

$$F_{zj}^3 = -a\Sigma \left[\frac{\partial^3 J_m(\beta\xi)}{\partial\xi^3} + \frac{1}{\xi}\frac{\partial^2 J_m(\beta\xi)}{\partial\xi^2} - \frac{m^2}{\xi^2}\frac{\partial J_m(\beta\xi)}{\partial\xi}\right] \cos(m\vartheta)e^{-ik_\eta\eta}, \tag{52}$$

$$F_{zj}^4 = -\Xi \left(\frac{k_p^2}{\rho\omega^2}\right) \frac{\partial G_{01}}{\partial\eta} \frac{1}{\alpha^2 - \kappa_z^2} \frac{\partial J_m(\kappa_z\xi)}{\partial\xi}. \tag{53}$$

Similarly, the displacement $d$ due to $P_z$ is expressed by

$$u_{dz} = P_z \left(A_{mz}F_{dz}^1 + B_{mz}F_{dz}^2 + C_{mz}F_{dz}^3 + F_{dz}^4\right) e^{-i\omega t}. \tag{54}$$

For the radial component $u_{rz}$,

$$F_{rz}^1 = -ik_\eta \frac{\partial J_m(\alpha\xi)}{\partial\xi} \cos(m\vartheta)\, e^{-ik_\eta\eta}, \tag{55}$$

$$F_{rz}^2 = F_{rz}^3 = 0, \tag{56}$$

$$F_{rz}^4 = -\left(\frac{k_p^2}{\rho\omega^2}\right) \frac{\partial G_{01}}{\partial\eta} \frac{1}{\alpha^2 - \kappa_z^2} \frac{\partial J_m(\kappa_z\xi)}{\partial\xi}. \tag{57}$$

For the tangential component $u_{\theta z}$,

$$F_{\theta z}^1 = \frac{ik_\eta m}{\xi} J_m(\alpha\xi) \sin(m\vartheta)e^{-ik_\eta\eta}, \tag{58}$$

$$F_{\theta z}^2 = F_{\theta z}^3 = F_{\theta z}^4 = 0. \tag{59}$$

For the axial component $u_{zz}$,

$$F_{zz}^1 = -k_\eta^2 J_m(\alpha\xi) \cos(m\vartheta)e^{-ik_\eta\eta}, \tag{60}$$

$$F_{zz}^2 = F_{zz}^3 = 0, \tag{61}$$

$$F_{zz}^4 = -\left(\frac{k_p^2}{\rho\omega^2}\right)\frac{\partial^2 G_{01}}{\partial\eta^2}\frac{1}{\alpha^2 - \kappa_z^2}\frac{\partial J_m(\kappa_z\xi)}{\partial\xi}. \tag{62}$$

As expressed by Equations (41) and (54), the displacement components involve the coupling constants $A_m$, $B_m$, and $C_m$. These constants can be determined directly by applying a fundamental set of linear elastic boundary problems. The outer and inner surfaces of the cylindrical shell studied in the present paper are stress-free. Thus, the following stress components are zero at $\xi_i$ and $\xi_j$, satisfying $\sqrt{(\xi_i + x_{0i})^2 + (\xi_i + x_{0i})^2} = a$ for the outer circumference and $\sqrt{(\xi_i + x_{0i})^2 + (\xi_i + x_{0i})^2} = b$ for the inner circumference in Equation (17) for the shell

$$\sigma_{rr} = \sigma_{r\theta} = \sigma_{rz} = 0. \tag{63}$$

In ref. [28], by using the stress displacement relations, we obtained a system of linear algebraic equations for the TIC, given by

$$\begin{bmatrix} a_{11f} & a_{12f} & a_{13f} \\ a_{21f} & a_{22f} & a_{23f} \\ a_{31f} & a_{32f} & a_{33f} \end{bmatrix}\begin{bmatrix} A_{mf} \\ B_{mf} \\ C_{mf} \end{bmatrix} = \begin{bmatrix} b_{1f} \\ b_{2f} \\ b_{3f} \end{bmatrix}, \tag{64}$$

where $f = j$ for $P_j$ and $f = z$ for $P_z$.

For $P_j$, all elements in Equation (64) are nonzero, as given by

$$a_{11j} = \Xi\left\{-\left(c_{12}\frac{m^2}{\xi^2} + c_{13}k_\eta^2\right)\frac{\partial J_m(\alpha\xi)}{\partial\xi} + c_{12}\frac{1}{\xi}\frac{\partial^2 J_m(\alpha\xi)}{\partial\xi^2} + c_{11}\frac{\partial^3 J_m(\alpha\xi)}{\partial\xi^3}\right\} \\ \times \cos(m\vartheta)\,e^{-ik_\eta\eta}, \tag{65}$$

$$a_{12j} = (-c_{11} + c_{12})\frac{m\Sigma}{\xi}\left[\frac{1}{\xi}\frac{\partial J_m(\beta\xi)}{\partial\xi} - \frac{\partial^2 J_m(\beta\xi)}{\partial\xi^2}\right]\cos(m\vartheta)e^{-ik_\eta\eta}, \tag{66}$$

$$a_{13j} = ik_\eta a\Sigma\left\{(c_{12} + c_{13})\frac{1}{\xi}\left[-\frac{m^2}{\xi}\frac{\partial J_m(\beta\xi)}{\partial\xi} + \frac{\partial^2 J_m(\beta\xi)}{\partial\xi^2}\right] + (c_{11} + c_{13})\frac{\partial^3 J_m(\beta\xi)}{\partial\xi^3}\right\} \\ \times \cos(m\vartheta)e^{-ik_\eta\eta}, \tag{67}$$

$$a_{21j} = \frac{(c_{11} - c_{12})}{2}(2m\Xi)\left[\frac{1}{\xi^2}\frac{\partial J_m(\alpha\xi)}{\partial\xi} - \frac{1}{\xi}\frac{\partial^2 J_m(\alpha\xi)}{\partial\xi^2}\right]\sin(m\vartheta)\,e^{-ik_\eta\eta}, \tag{68}$$

$$a_{22j} = -\frac{(c_{11} - c_{12})}{2}\Sigma\left[\frac{m^2}{\xi}\frac{\partial J_m(\beta\xi)}{\partial\xi} + \left(2 - \frac{1}{\xi}\right)\frac{\partial^2 J_m(\beta\xi)}{\partial\xi^2} + \frac{\partial^3 J_m(\beta\xi)}{\partial\xi^3}\right]\sin(m\vartheta)e^{-ik_\eta\eta}, \tag{69}$$

$$a_{23j} = (c_{11} - c_{12})\left(\frac{ik_\eta am\Sigma}{\xi}\right)\left[\frac{1}{\xi}\frac{\partial J_m(\beta\xi)}{\partial\xi} - \frac{\partial^2 J_m(\beta\xi)}{\partial\xi^2}\right]\sin(m\vartheta)e^{-ik_\eta\eta}, \tag{70}$$

$$a_{31j} = -2c_{44}(ik_\eta\Xi)\frac{\partial^2 J_m(\alpha\xi)}{\partial\xi^2}\cos(m\vartheta)\,e^{-ik_\eta\eta}, \tag{71}$$

$$a_{32j} = -c_{44}\left(\frac{ik_\eta m\Sigma}{\xi}\right)\frac{\partial J_m(\beta\xi)}{\partial\xi}\cos(m\vartheta)e^{-ik_\eta\eta}, \tag{72}$$

$$a_{33j} = c_{44}a\Sigma\left[-\frac{2m^2}{\xi^3}\frac{\partial J_m(\beta\xi)}{\partial\xi} + \left(\frac{1+m^2}{\xi^2} + k_\eta^2\right)\frac{\partial^2 J_m(\beta\xi)}{\partial\xi^2} - \frac{1}{\xi}\frac{\partial^3 J_m(\beta\xi)}{\partial\xi^3} \\ - \frac{\partial^4 J_m(\beta\xi)}{\partial\xi^4}\right]\cos(m\vartheta)\,e^{-ik_\eta z}, \tag{73}$$

$$b_{1j} = -\left(\frac{k_p^2}{\rho\omega^2}\right)\frac{\Xi}{\alpha^2 - \kappa_z^2}\left[c_{11}G_{01}\frac{\partial^3 J_m(\kappa_z\xi)}{\partial\xi^3} + c_{12}\frac{G_{01}}{\xi}\frac{\partial^2 J_m(\kappa_z\xi)}{\partial\xi^2} + c_{13}\frac{\partial^2 G_{01}}{\partial\eta^2}\frac{\partial J_m(\kappa_z\xi)}{\partial\xi}\right], \tag{74}$$

$$b_{2j} = \frac{(c_{11} - c_{12})}{2}\left(\frac{k_s^2}{\rho\omega^2}\right)\Sigma\,G_{01}\frac{1}{\beta^2 - \kappa_z^2}\left[\frac{1}{\xi}\frac{\partial^2 J_m(\kappa_z\xi)}{\partial\xi^2} - \frac{\partial^3 J_m(\kappa_z\xi)}{\partial\xi^3}\right], \tag{75}$$

$$b_{3j} = -c_{44} \left( \frac{k_p^2}{\rho\omega^2} \right) \frac{2\Xi}{\alpha^2 - \kappa_z^2} \frac{\partial G_{01}}{\partial\eta} \frac{\partial^2 J_m(\kappa_z\xi)}{\partial\xi^2}, \tag{76}$$

For $P_z$,

$$a_{11z} = ik_\eta \left[ \left( c_{12}\frac{m^2}{\xi^2} + c_{13}k_\eta^2 \right) J_m(\alpha\xi) - c_{12}\frac{\partial J_m(\alpha\xi)}{\partial\xi} - c_{11}\frac{\partial^2 J_m(\alpha\xi)}{\partial\xi^2} \right] \cos(m\vartheta)\, e^{-ik_\eta\eta}, \tag{77}$$

$$a_{12z} = a_{13z} = 0, \tag{78}$$

$$a_{21z} = (c_{11} - c_{12})(ik_\eta m) \left[ -\frac{1}{\xi^2}J_m(\alpha\xi) + \frac{1}{\xi}\frac{\partial J_m(\alpha\xi)}{\partial\xi} \right] \sin(m\vartheta)\, e^{-ik_\eta\eta}, \tag{79}$$

$$a_{22z} = a_{23z} = 0, \tag{80}$$

$$a_{31z} = -2c_{44}k_\eta^2 \frac{\partial J_m(\alpha\xi)}{\partial\xi} \cos(m\vartheta)\, e^{-ik_\eta\eta}, \tag{81}$$

$$a_{32z} = a_{33z} = 0, \tag{82}$$

$$b_{1z} = -\left( \frac{k_p^2}{\rho\omega^2} \right) \frac{1}{\alpha^2 - \kappa_z^2} \left[ c_{11}\frac{\partial G_{01}}{\partial\eta} \frac{\partial^2 J_m(\kappa_z\xi)}{\partial\xi^2} + c_{12}\frac{1}{\xi}\frac{\partial G_{01}}{\partial\eta} \frac{\partial J_m(\kappa_z\xi)}{\partial\xi} \right.$$
$$\left. + c_{13}\frac{\partial^3 G_{01}}{\partial\eta^3} \frac{\partial J_m(\kappa_z\xi)}{\partial\xi} \right], \tag{83}$$

$$b_{2z} = 0, \tag{84}$$

$$b_{3z} = -c_{44} \left( \frac{k_p^2}{\rho\omega^2} \right) \frac{\partial^2 G_{01}}{\partial\eta^2} \frac{1}{\alpha^2 - \kappa_z^2} \left[ \frac{\partial^2 J_m(\kappa_z\xi)}{\partial\xi^2} + \frac{\partial J_m(\kappa_z\xi)}{\partial\xi} \right], \tag{85}$$

For $P_z$, by substituting Equations (77)–(85) into Equation (64), we derive

$$A_{mz} = \frac{b_{1z}}{a_{11z}} = \frac{b_{3z}}{a_{31z}}. \tag{86}$$

Equation (86) allows us to obtain the value of $\kappa_z$ at a given point on the circumference. The only remaining task to complete the displacement fields is to introduce retardation times into the CF $\boldsymbol{P}$ in Equation (1). The CF is an impulsive force acting at the PS-oriented origin $\xi_i = 0$, $\xi_j = 0$, and $\eta = 0$ at $t = 0$. The arrival time $\tau$ of the signal at position $(\xi_i, \xi_j, \eta)$ must be considered. The arrival times of the P and S waves propagating with velocities $c_P$ and $c_S$ are given as

$$\tau_P = \frac{\sqrt{\varsigma^2 + \eta^2}}{c_P}, \ \tau_S = \frac{\sqrt{\varsigma^2 + \eta^2}}{c_s}, \tag{87}$$

respectively, where $\varsigma$ is given by Equation (18). Introducing $P_0$ and $b$ parameters, determining the amplitude and duration of the wave, respectively, yields the CF acting in the $f$ direction as

$$\boldsymbol{P} = P_{0f}(t - \tau)e^{-b(t-\tau)}. \tag{88}$$

In the simulation, $P_0 = 1.0 \times 10^{10}$ N s$^{-1}$ and $b = 1.0 \times 10^{-5}$ s$^{-1}$.
Finally, the displacements generated by $P_f$ can be obtained as

$$u_{rf} = P_f \left[ (t - \tau_P)\left( A_{mf}F_{rf}^1 + F_{rf}^4 \right) + (t - \tau_S)\left( B_{mf}F_{rf}^2 + C_{mf}F_{rf}^3 \right) \right] e^{-i\omega t}, \tag{89}$$

$$u_{\theta f} = P_f \left[ (t - \tau_P)\left( A_{mf}F_{\theta f}^1 \right) + (t - \tau_S)\left( B_{mf}F_{\theta f}^2 + C_{mf}F_{\theta f}^3 + F_{\theta f}^4 \right) \right] e^{-i\omega t}, \tag{90}$$

$$u_{zf} = P_f \left[ (t - \tau_P)\left( A_{mf}F_{zf}^1 + F_{zf}^4 \right) + (t - \tau_S)\left( B_{mf}F_{zf}^2 \right) \right] e^{-i\omega t}. \tag{91}$$

For practical purposes, the P, SH, and SV waves are introduced as

$$u_f^P = P_f(t - \tau_P)\left( A_{mf}F_{df}^1 + F_{rf}^4 \right) e^{-i\omega t} \ \rightarrow (d = r, z \text{ and } \theta), \tag{92}$$

$$u_f^{SV} = P_f(t - \tau_S)\left(C_{mf}F_{df}^3\right)e^{-i\omega t} \rightarrow\rightarrow (d = r, z \text{ and } \theta), \tag{93}$$

$$u_f^{SH} = P_f(t - \tau_S)\left(B_{mf}F_{df}^2 + F_{\theta f}^4\right)e^{-i\omega t} \rightarrow (d = r, z \text{ and } \theta). \tag{94}$$

## 4. Simulations

In ref. [28], simulations of the displacement fields in TIC were confined to the case for the azimuthal independence ($m = 0$) of wave propagation. In this paper, we extended the simulation to the case of azimuthally dependent tangential displacements, i.e., $2\pi$-aperiodic solutions. Stainless steel cylindrical structures ($a = 0.50$ m, $l = 2.0$ m, $\rho = 7.80 \times 10^3$ kg/m$^3$, $c_P = 5.98$ km/s, $c_s = 3.30$ km/s, and $\nu = 155.4$ kHz) were used as the test specimens. First, we determined $k_\eta$ from Equation (86), resulting in two solutions of $A_{mz}$.

$$A_{mz}^1 = \left.\frac{b_{1z}}{a_{11z}}\right|_{\xi=\varsigma} \cdot \text{or} \rightarrow A_{mz}^2 = \left.\frac{b_{3z}}{a_{31z}}\right|_{\xi=\varsigma}. \tag{95}$$

On the outer circumference, $k_z = \frac{r_{v1}}{(a-r_0)}$, the first root of the function

$$f(\eta) = \left.\left(\frac{b_{1z}}{a_{11z}} - \frac{b_{3z}}{a_{31z}}\right)\right|_{\xi=\varsigma_o} = 0, \tag{96}$$

Was solved at a given PS location as a function of $\eta$ ($= l - z_0$) and the shortest distance $\varsigma_o$ from the PS to a given point (0.5 m, 45°, $\eta$) on the outer circumference of the cylindrical structures. Figure 3 shows the dependence of $k_\eta / \pi$ on $\eta$ for the cylinder ($b = 0$) and shell ($b = 0.4$ m). As shown in the figure, the $k_\eta$ values are independent of the inner diameter $b$. When PS is located on a radial axis, the dependence of the $\eta$-dependency of $k_\eta$ values is very simple: all values are divided into two groups with even $m$ and odd $m$. It should be noted that the $k_\eta$ values are almost independent of $\vartheta$.

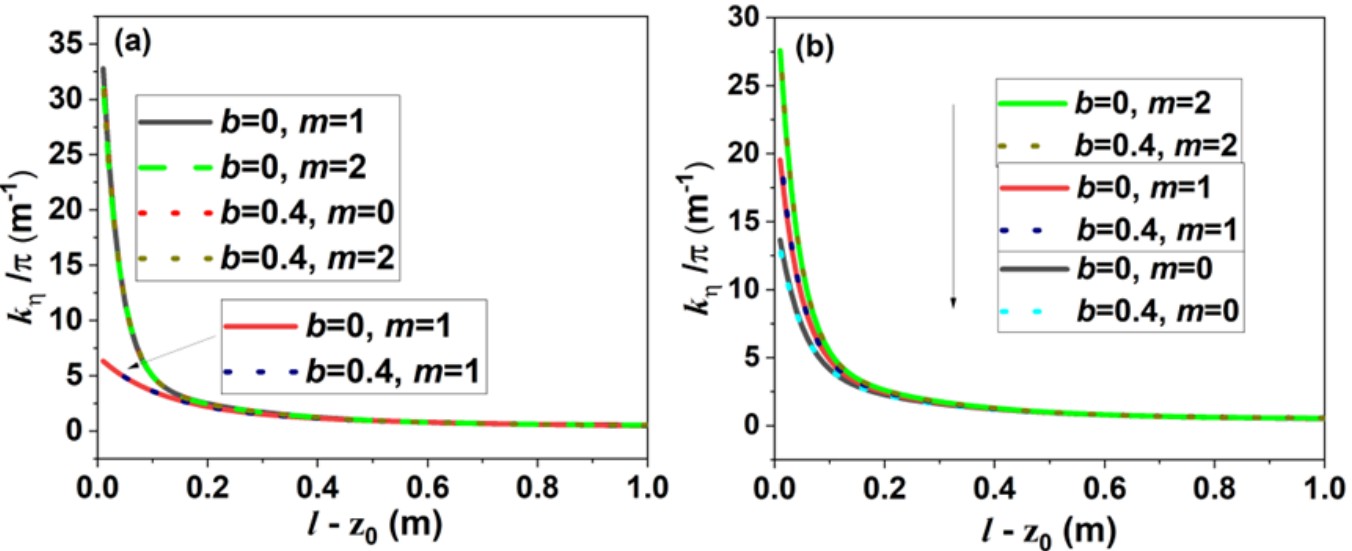

**Figure 3.** Spectra of $k_\eta$ values calculated as a function of $\eta$ ($l-z_0$): (**a**) $x_{0i} = 0$, $x_{0j} = 0.45$, $z_0 = 1$ m and (**b**) $x_{0i} = 0.35$, $x_{0j} = 0.35$, $z_0 = 1$ m.

Using these $k_\eta$ results, we simulated the displacement fields at the outer surfaces of the cylinder ($b = 0$ m) and cylindrical shell ($b = 0.4$ m). Figure 4 shows the $2\pi$-periodic ($m = 0$) displacements and their wave properties at the (0.5 m, 45°, 1 m) position, generated by the $P_f$ PS located at the center of the cylinder $\left(x_{0i} = 0, x_{0j} = 0, z_0 = 1 \text{ m}\right)$. The $P_j$ and $P_z$ excitations produce an axial displacement stronger than the radial and tangential displacements. For the $P_j$ excitation, the displacement results in the P wave are the main wave,

with a minor SH wave and very weak SV wave. In Figure 4, the displacement amplitudes generated by $P_z$ excitation differ significantly from those generated by the $P_j$ excitation, in which the $P_z$ excitation produces only the P wave. For the $P_j$ excitation, the maximum values of $u_{rj}$ and $u_{zj}$ at the (0.5 m, 45°, 1 m) position were 6.7 and 17.4 nm, respectively, while for the $P_z$ excitation they were 22.1 and 127.0 nm, respectively. The amplitudes of the displacements due to $P_z$ excitation are much stronger than those due to the $P_j$ excitation.

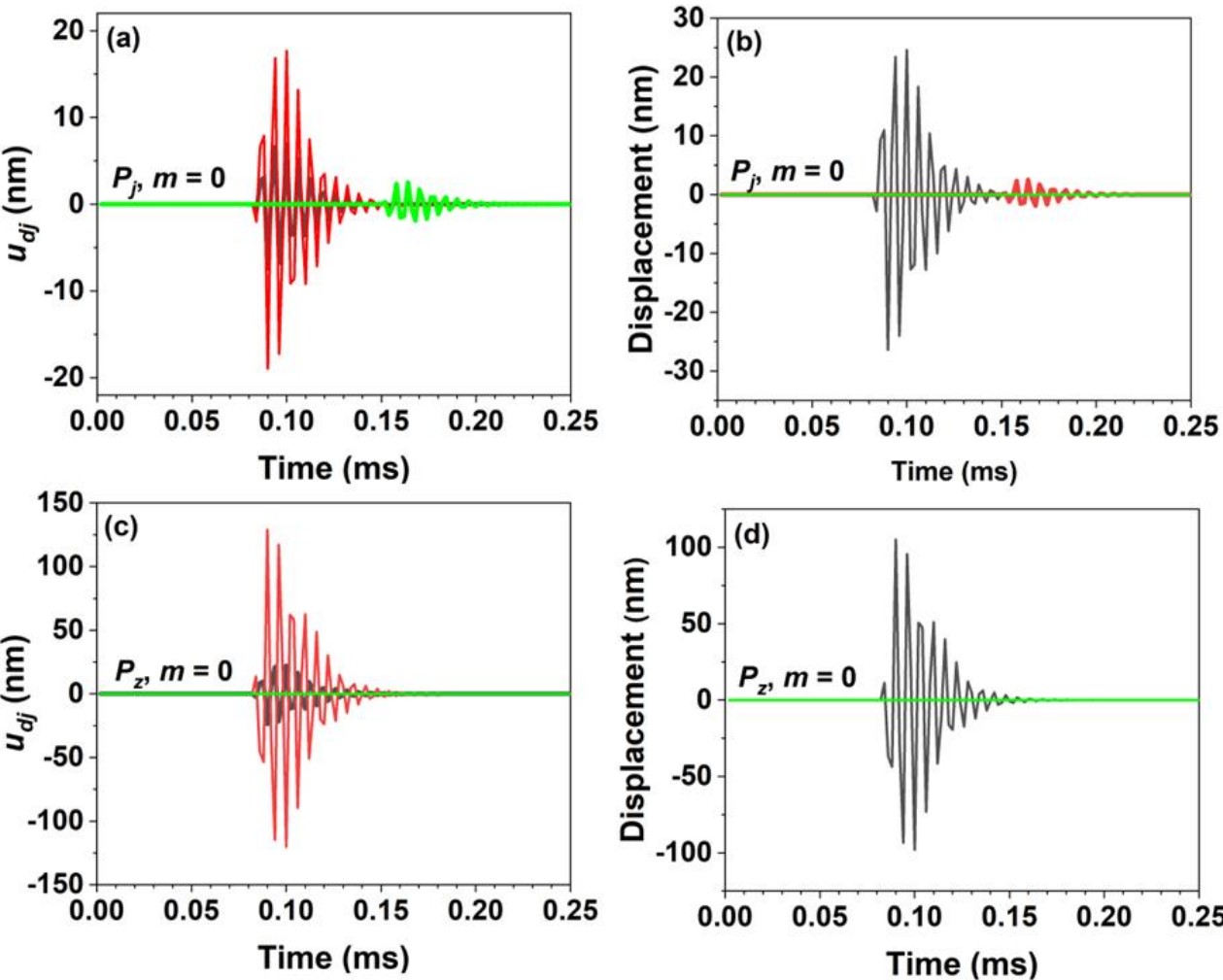

**Figure 4.** The $2\pi$-periodic ($m = 0$) displacements at the (0.5 m, 45°, 1 m) position generated by the PS of 1.0 N located at the center of the cylinder ($P_j$: upper, $P_z$: lower): (**a,c**) radial (black line), axial (red line), and tangential (green line) components; and (**b,d**) P (black line), SH (red line), and SV (green line) waves.

The angular dependence of the displacement was also calculated as a function of $m$ ($m = 0$, 1, and 2), as shown in Figure 5. For $m = 0$, when the PS is located at the center of the circular plane, the angular dependences of $u_{rj}$, $u_{zj}$, and $u_{tj}$ arise only from $\Xi$ and $\Sigma$, defined in Equations (32) and (35), respectively. For $P_z$ excitation, the displacements of $u_{rz}$ and $u_{zz}$ are free from these factors. When the distances from the PS to the circumference are not equivalent, the angular dependences of the radial and axial displacements are highly significant. These effects are due not only to $\Xi$ and $\Sigma$, but also to the superposition of the Bessel functions involved in the displacement Equations. When $m$ is nonzero, additional azimuthal factors, $m$, $\cos(m\vartheta)$, and $\sin(m\vartheta)$, result in complex angular dependency. It should be noted that at a certain angle, some $a_{ij}$ values of $u_{rj}$, $u_{zj}$, and $u_{rz}$ in Equation (64) become too small to cause the sudden increase in displacement, hereafter referred to as "computational divergency". Figure 6 shows the angular dependence of a relatively

strong $u_{zz}$ generated by two PSs located at $x_{0i} = 0$, $x_{0j} = 0.45$ m and $z_0 = 1$ m, and $x_{0i} = 0.35$ m, $x_{0j} = 0.35$ m and $z_0 = 1$ m. For the first PS, the distance from the PS to the point at $\xi_i = 0$, $\xi_j = 0.5$ m, and $\eta = 0$ on the outer surface is the shortest with $\varsigma = 0.05$ m.

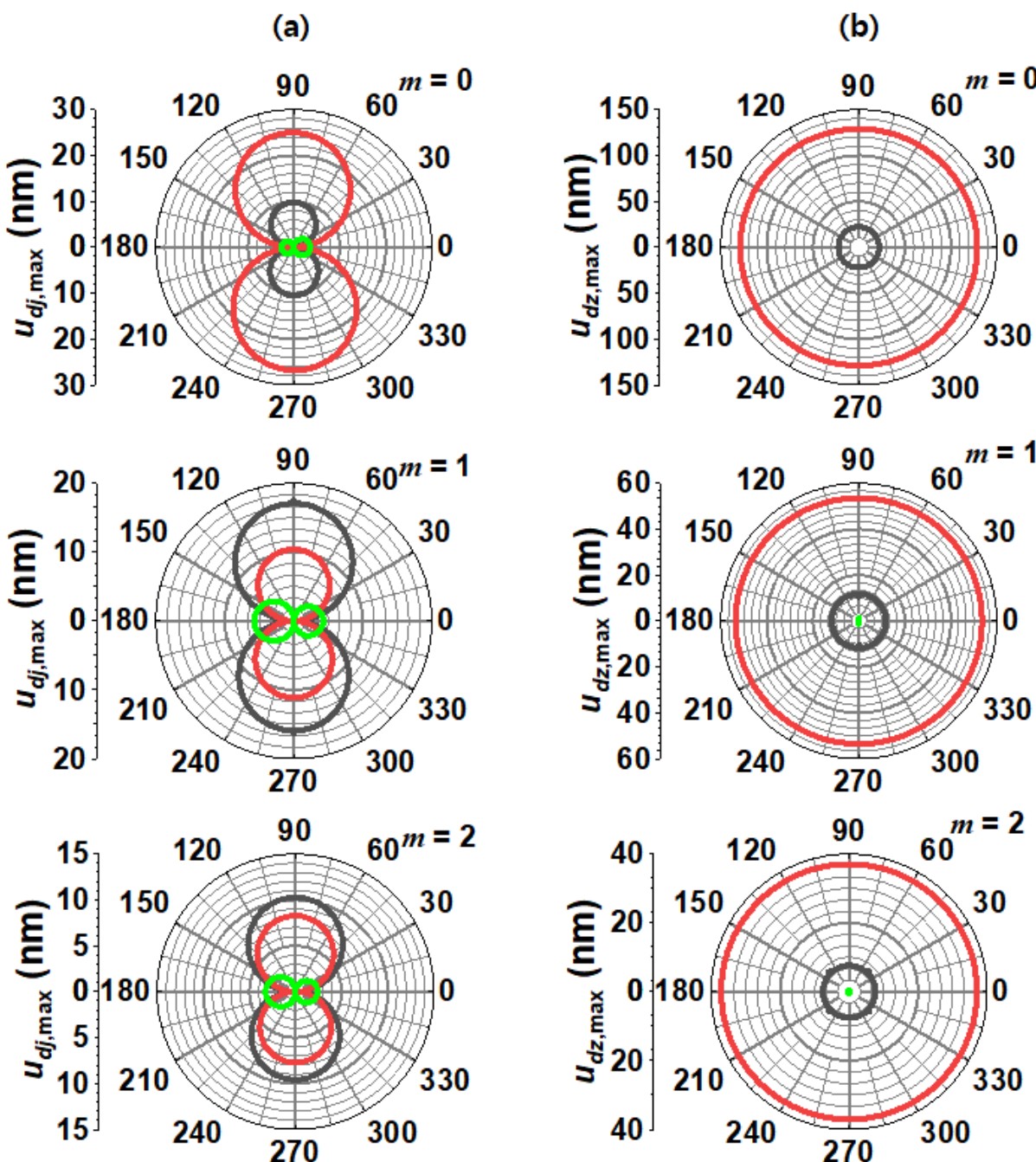

**Figure 5.** Angular dependences of the maximum displacements on the circumference, with $r = 0.5$ m and $z = 1$ m for the cylinder: radial (black line), axial (red line), and tangential (green line) displacements generated by (**a**) $P_j$ and (**b**) $P_z$ of 1.0 N, located at $x_{0i} = 0$, $x_{0j} = 0$, and $z_0 = 1$ m.

For this case, $\tan^{-1}\left(\frac{\xi_j}{\bar{\xi}_j}\right) = 90°$. For the second PS, the distance along $\tan^{-1}\left(\frac{\xi_j}{\bar{\xi}_j}\right) = 45°$ is the shortest from PS ($\varsigma = 0.005$ m). Spectra of the angular dependences of the displacement fields are symmetric with respect to $\theta = \tan^{-1}\left(\frac{\xi_j}{\bar{\xi}_j}\right)$. As shown in Figure 6, the $u_{zz}$ displacements at a point very close to the PS ($\varsigma = 0.005$ m) are extremely large.

Remarkably, $P_z$ excitation of 1 N produces a P wave with a maximum amplitude of tens of centimeters.

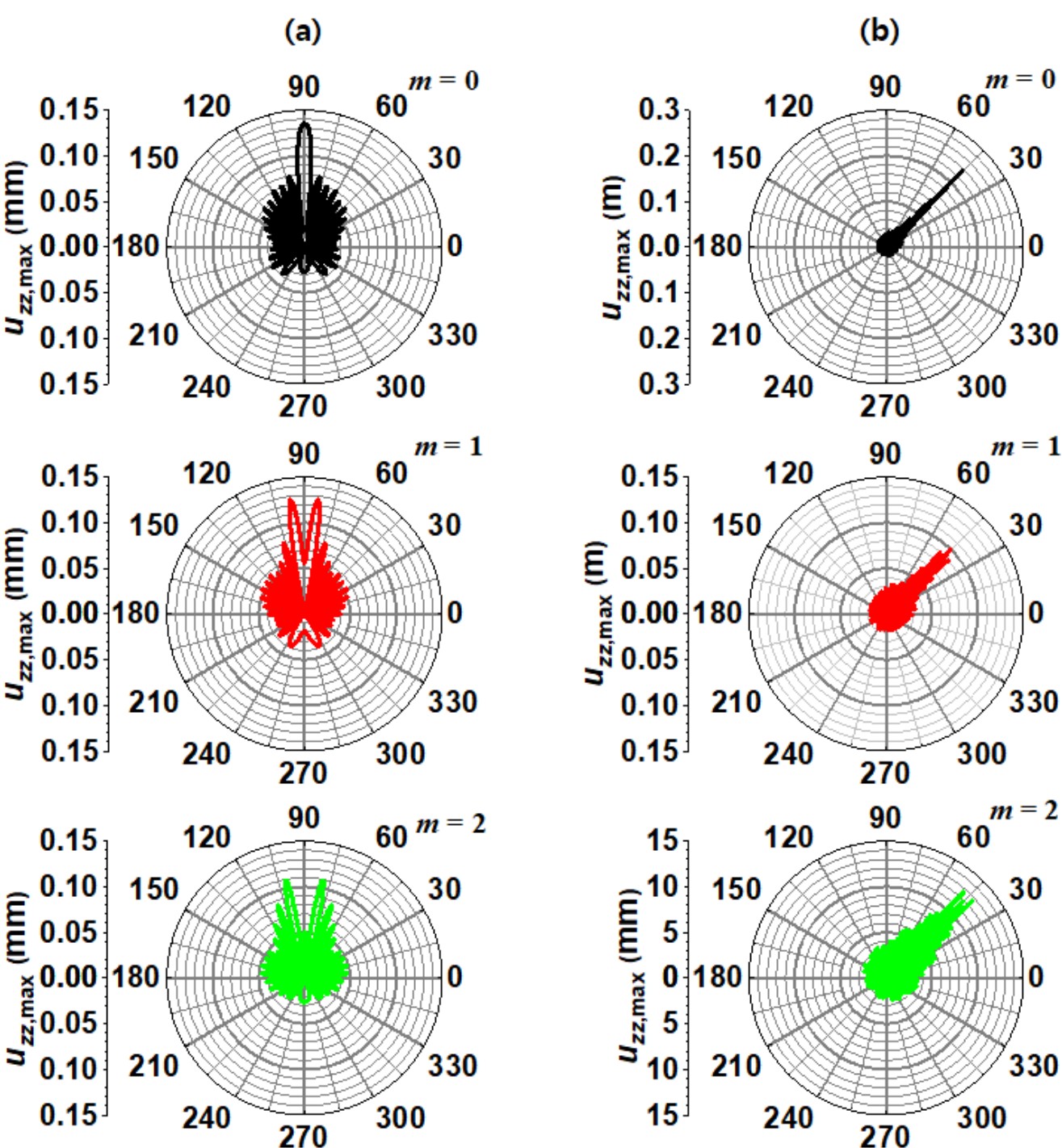

**Figure 6.** Angular dependences of the maximum displacements of $u_{zz}$ on the circumference at $z = 1$ m for the cylinder ($a = 0.5$ m, $b = 0$ m), generated by the PS located at (**a**) (0 m, 0.45 m, 1 m), and (**b**) (0.35 m, 0.35 m, 1 m).

Similarly, the displacement fields and their wave properties excited by the PS were also calculated as a function of $m$ ($m = 0$, 1, and 2) for the shell ($a = 0.5$ m, $b = 0.4$ m). Figure 7 shows the angular dependences of the axial component $u_{zz}$ generated by two PSs located at $x_{0i} = 0$, $x_{0j} = 0.45$ m and $z_0 = 1$ m, and $x_{0i} = x_{0j} = 0.35$ m and $z_0 = 1$ m. There is no difference in the spectral features of the corresponding angular dependences between the

cylinder and shell geometries. However, the maximum amplitudes of the shell are much smaller than those of the cylinder. Figure 8 shows $2\pi$-periodic ($m = 0$) displacements at the (0.5 m, 45°, 1 m) position. For the shell, the most striking feature of the displacement fields is the tangential displacement $u_{tj}$ due to $P_j$ excitation, the amplitude of which is comparable to that of the axial displacement $u_{zj}$, as shown in Figure 8a.

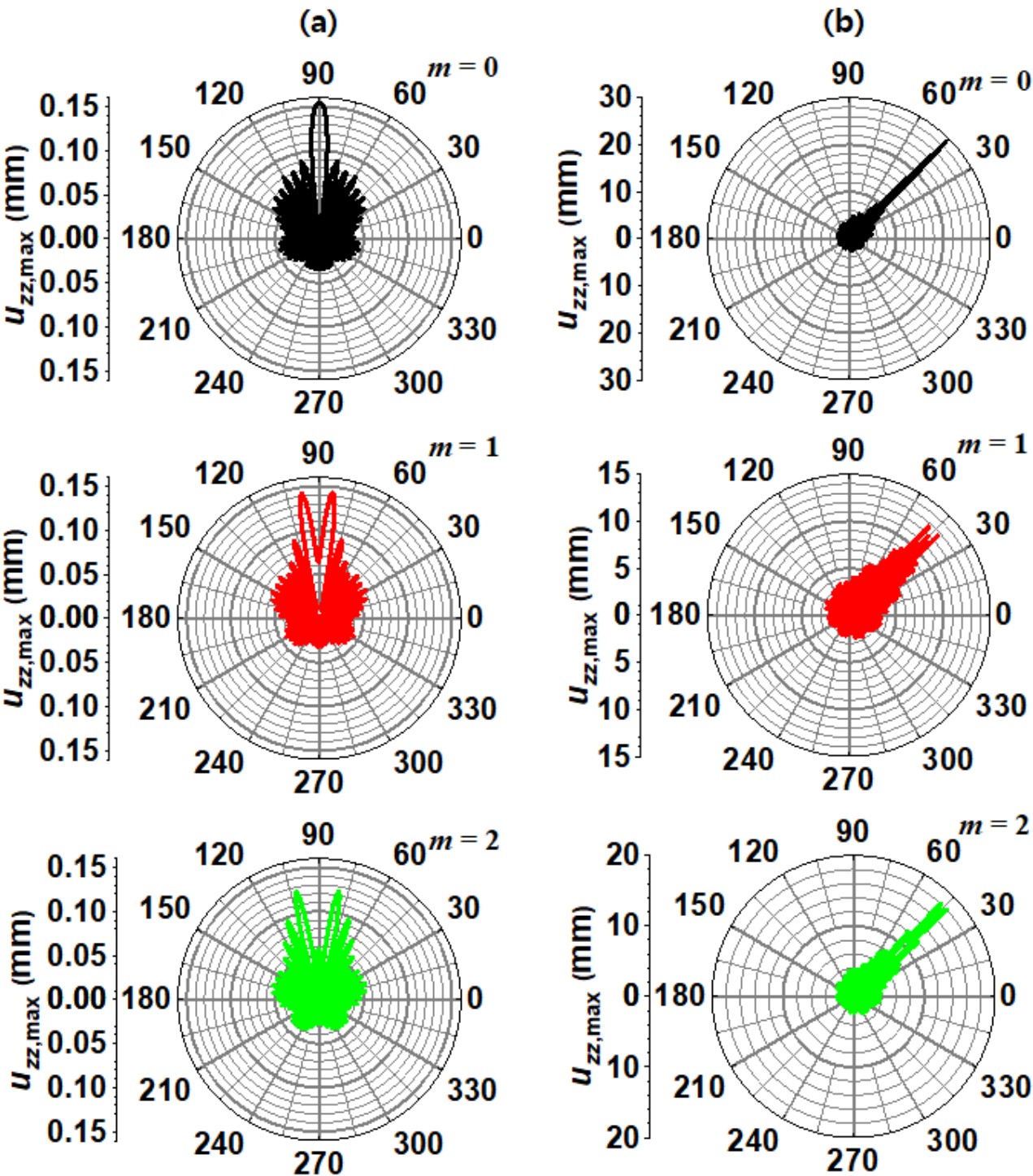

**Figure 7.** Angular dependences of the maximum displacements of $u_{zz}$ on the circumference at $z = 1$ m for the shell ($a = 0.5$ m, $b = 0.4$ m), generated by the PS located at: (**a**) (0 m, 0.45 m, 1 m), and (**b**) (0.35 m, 0.35 m, 1 m).

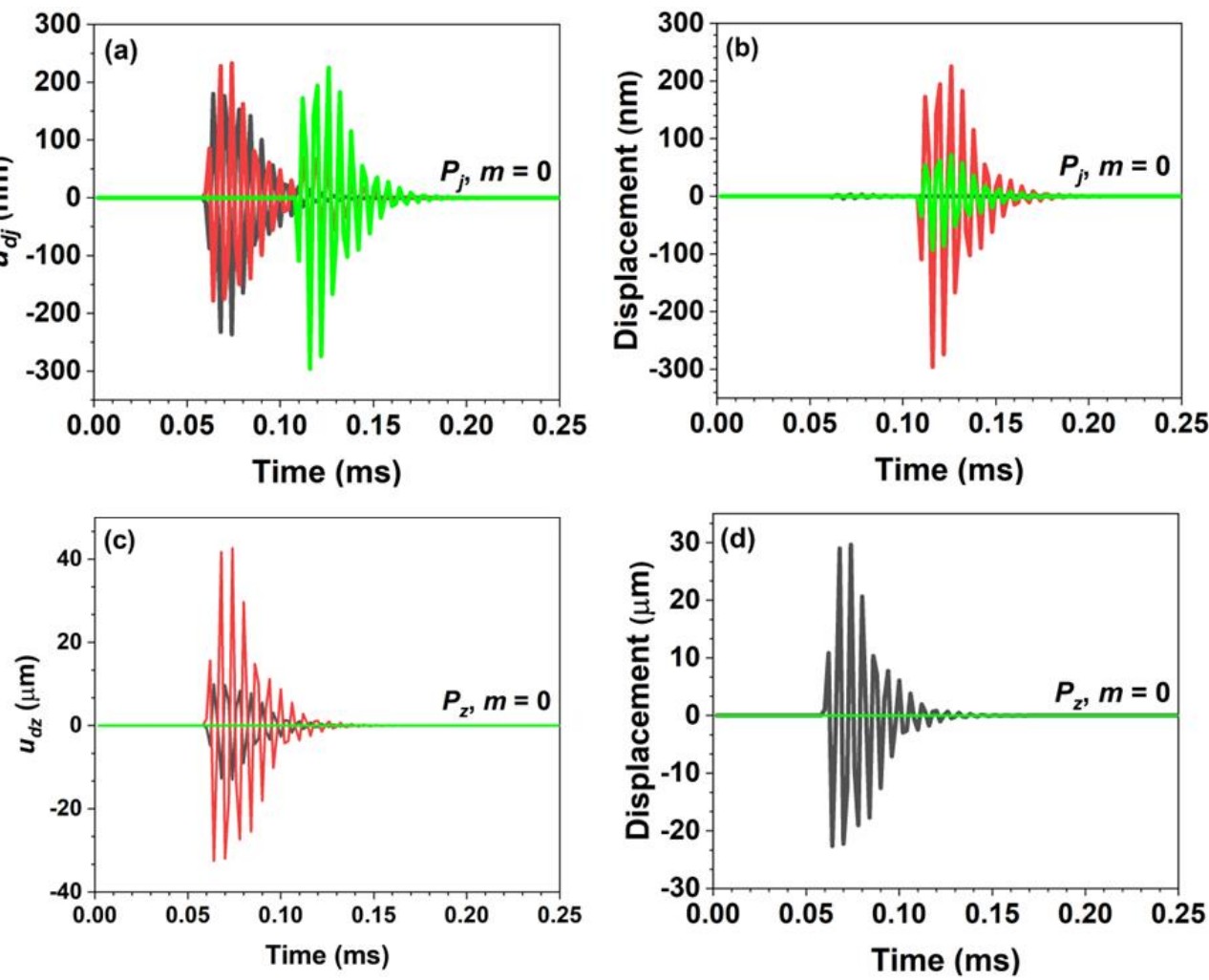

**Figure 8.** The 2π-periodic displacements at the (0.5 m, 45°, 1 m) position generated by the PS of 1.0 N located at (0 m, 0.45 m, 1 m) on the cylindrical shell ($P_j$: upper, $P_z$: lower): (**a**,**c**) radial (black line), axial (red line), and tangential (green line) components; and (**b**,**d**) P (black line), SH (red line), and SV (green line) waves.

## 5. Conclusions

In this paper, we provided a mathematical model for AE generated by an intrinsic PS in cylindrical geometries, including a cylinder and shell. As an internal crack, the PS produces CFIPs for P, SH, and SV waves. Introducing CFIPs into the NL equation provides solutions for the radial, tangential, and axial displacements, involving azimuthal functions in cylindrical geometries. The main advantage of our model is that it provides an exact solution for the AE features from PS generation, propagation, and reception in cylindrical geometries. In conjunction with experimental data, this mathematical model can be used for NDT of cylindrical structures.

**Author Contributions:** Conceptualization, methodology and investigation, K.B.K. and J.-G.K.; software, validation, and formal analysis, K.B.K.; resources and data curation, K.B.K.; writing—original draft preparation, J.-G.K.; writing—review and editing, visualization, and supervision, J.-G.K.; project administration and funding acquisition, B.K.K. All authors have read and agreed to the published version of the manuscript.

**Funding:** This research was funded by the Korea Institute of Energy Technology Evaluation and Planning (KETEP) and the Ministry of Trade, Industry & Energy (MOTIE) of the Republic of Korea (NO. 20202910100070).

**Institutional Review Board Statement:** Not applicable.

**Informed Consent Statement:** Not applicable.

**Data Availability Statement:** Not applicable.

**Conflicts of Interest:** The authors declare no conflict of interest.

## Appendix A

Erratum: Kim, K.B.; Kim, B.K.; Lee, S.G.; Kang, J.-G. Analytical Modeling of Acoustic Emission Due to an Internal Point Source in a Transversely Isotropic Cylinder [28].

The integration range on the left side of Equation (34) should apply from 0 to *a*. Therefore, Equation (36) should be read

$$A_{v1} = \frac{1}{\pi \left[ e^{\kappa_z z_0} + e^{-\kappa_z (l - z_0)} - 2 \right]} \times \frac{2}{a^2 [J_{v+1}(a\kappa_z)]^2} \quad \left( \kappa_z = \frac{r_{v1}}{a - r_0} \right). \tag{A1}$$

PDE for $\phi_r$ is given by

$$\frac{\partial^2 \phi_\xi}{\partial \xi^2} + \frac{1}{\xi} \frac{\partial \phi_\xi}{\partial \xi} + \left( \alpha^2 - \frac{m^2}{\xi^2} \right) \phi_\xi = -\frac{k_p^2}{\rho \omega^2} \left( \frac{1}{\phi_\vartheta \phi_\eta} \right) G_1 J_0 \left( \frac{r_{01}}{a} \xi \right).$$

The solution of this PDE is a linear combination of the homogeneous ($\phi_{\xi h}$) and particular ($\phi_{\xi p}$) solutions. For an inhomogeneous equation,

$$\frac{d^2 u}{dz^2} + \frac{1}{z} \frac{du}{dz} + \left( 1 - \frac{v^2}{z^2} \right) u = Z_\mu(\lambda z),$$

The Korenev's particular solution is given by [32]

$$u = \frac{Z_v(\lambda z)}{1 - \lambda^2}, \lambda \neq 1.$$

After applying this solution to PDE of $\phi_r$, Equations (66)–(68) should be read

$$\phi_{\xi p} = -\frac{k_p^2}{\rho \omega^2} \left( \frac{1}{\phi_\vartheta \phi_\eta} \right) G_1(\eta) \frac{1}{\alpha^2 - \left( \frac{r_{01}}{a - r_0} \right)^2} J_m \left( \frac{r_{01}}{a - r_0} \xi \right) \quad \frac{r_{01}}{\alpha(a - r_0)} \neq 1, \tag{A2}$$

$$\begin{aligned} \phi(\xi, \vartheta, \eta) &= \phi_r(\xi) \phi_\theta(\vartheta) \phi_z(\eta) \\ &= [A_{mr} J_m(\alpha \xi)] [A_{m\theta} cos(m\vartheta)] \left( A_z e^{-ik_\eta \eta} \right) - \frac{k_p^2}{\rho \omega^2} G_1(\eta) \frac{1}{\alpha^2 - \left( \frac{r_{01}}{a - r_0} \right)^2} J_m \left( \frac{r_{01}}{a - r_0} \xi \right), \end{aligned} \tag{A3}$$

$$\chi(\xi, \vartheta, \eta) = B_m J_m(\beta \xi) \sin(m\vartheta) e^{-ik_\eta \eta} - \frac{k_p^2}{\rho \omega^2} G_1(\eta) \frac{1}{\beta^2 - \left( \frac{r_{01}}{a - r_0} \right)^2} J_m \left( \frac{r_{01}}{a - r_0} \xi \right), \tag{A4}$$

Respectively.

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
