# Peer review of "Modeling Acoustic Emission Due to an Internal Point Source in Circular Cylindrical Structures"

_applsci, doi:10.3390/app122312032_

Round 1
Reviewer 1 Report
1. IntroductionExtensive and well-executed research of a theoretical nature. An unquestionable strength
of the work is the meticulous derivation of the individual dependencies of the
components of the displacement field: the radial, tangential and axial involving azimuthal
functions for the circular cylindrical structures. The introduction is brief, clear and comprehensive.
2. Comments
a) remarks on the content of the research
no comments
b) remarks on the graphs
- on Fig. 1 the coordinate system xi, xj for cross section of the cylindrical shell is presented and the results of displacements on Fig. 4 and Fig. 5 are also related to the two-dimensional reference system. So, why the comment (in line 96) about the z-direction axis? As a practice-scientist, I have some difficulty in correctly locating where the forcing force is applied or where the defect is simulated especially along the longitudinal axis (L-dimension). Figure 1 and its description are not sufficient to correctly understand the authors' intent. Perhaps an additional drawing in Chapter 4 would be useful?
c) minor remarks
I attach the detailed comments below:
- line 26: “…testing loadings are applied before and/or during or after the testing…”
- line37: instead of the word “motoring” it should be monitoring
- line 72: “The exact solution were solved and applied…” please rephrase this sentence
- line 208 and other: I do not understand the meaning of the word circumstance in the context of the problems under consideration
- line 229: should be: Simulations or Analytical simulations
- line 238: something went wrong with the formulation of the eta variable
d) others comments
no comments
3. Summary and suggestions for possible further research
If the authors manage to carry out numerical simulations with the use of FEM programs,
which will confirm the assumptions made and the obtained solution - it will be a great success in the area of non-destructive methods. The next step is laboratory testing of samples and later also inspections of actual structures. Then we will get to know the strength, advantages and probably various limitations of the mathematical apparatus formulated by the authors. I wish the authors of this article continued research success.
Author Response
- Introduction
Extensive and well-executed research of a theoretical nature. An unquestionable strength
of the work is the meticulous derivation of the individual dependencies of the
components of the displacement field: the radial, tangential and axial involving azimuthal
functions for the circular cylindrical structures. The introduction is brief, clear and comprehensive.
- Comments
- a) remarks on the content of the research
no comments
- b) remarks on the graphs
- on Fig. 1 the coordinate system xi, xj for cross section of the cylindrical shell is presented and the results of displacements on Fig. 4 and Fig. 5 are also related to the two-dimensional reference system. So, why the comment (in line 96) about the z-direction axis? As a practice-scientist, I have some difficulty in correctly locating where the forcing force is applied or where the defect is simulated especially along the longitudinal axis (L-dimension). Figure 1 and its description are not sufficient to correctly understand the authors' intent. Perhaps an additional drawing in Chapter 4 would be useful?
Þ Figure 1 is redrawn, in which two forms of the PS vector along the xj and z directions is supplemented.
- c) minor remarks
I attach the detailed comments below:
- line 26: “…testing loadings are applied before and/or during or after the testing…”
- line37: instead of the word “motoring” it should be monitoring
- line 72: “The exact solution were solved and applied…” please rephrase this sentence
- line 208 and other: I do not understand the meaning of the word circumstance in the context of the problems under consideration
- line 229: should be: Simulations or Analytical simulations
- line 238: something went wrong with the formulation of the eta variable
Þ All comments are supplemented in the revised paper.
- d) others comments
no comments
- Summary and suggestions for possible further research
If the authors manage to carry out numerical simulations with the use of FEM programs,
which will confirm the assumptions made and the obtained solution - it will be a great success in the area of non-destructive methods. The next step is laboratory testing of samples and later also inspections of actual structures. Then we will get to know the strength, advantages and probably various limitations of the mathematical apparatus formulated by the authors. I wish the authors of this article continued research success.
Þ Thank you very much for your suggestions for further research. As further work, we will conduct the stress-strain testing for cylindrical rod and shell to which the proposed theoretical models are applied.
Reviewer 2 Report
Manuscript ID: applsci-2023486
Title: Modeling Acoustic Emission due to an Internal Point Source in Circular Cylindrical Structures
In this paper, a mathematical model for acoustic emission generated by an intrinsic point source in circular cylindrical geometries, including a cylinder and shell, is presented. The paper is well written. The following minor comments should be addressed:
• The quality of the figures should be improved. (Figures 3-8)
• Explain the differences between this paper with the previous one (Appl. Sci. 2022, 12, 5272) in more detail.
• The conclusion should be enriched.
• Lattice Boltzmann method is a suitable method to simulate acoustic emissions. It is suggested to introduce this method for reader information in the introduction. Referring to the following paper will be useful: 10.1016/j.camwa.2022.07.005
Reviewer 3 Report
I have attached file to this email

Round 2
Reviewer 3 Report
Some of the corrections have not been made and the article cannot be published.
Author Response
The English in this document has been checked by at least two professional editors, both native speakers of English. For a certificate, please see: http://www.textcheck.com/certificate/Vnfqf2